Microbiology **Spectrum**

# The diagnostic accuracy of the GeneXpert ESBL-*ampC* prototype assay for rapid PCR-based detection of extended-spectrum beta-lactamase genes directly from urine

Sofie C. M. Tops,[1] Claire E. P. Schapendonk,[1] Jordy P. M. Coolen,[1] Fred C. Tenover,[2] Isabella A. Tickler,[3] Willem J. G. Melchers,[1] Heiman F. L. Wertheim[1]

**ABSTRACT**   The emerging prevalence of extended-spectrum beta-lactamase (ESBL) producing Enterobacterales has implications for the empirical treatment of common infections, such as complicated urinary tract infections. To provide adequate treatment, while avoiding empirical therapy with last resort carbapenems, rapid identification of ESBL-containing pathogens is desirable. Routine urine samples were collected between February and July 2021 in two Dutch clinical medical microbiology laboratories according to a predefined list containing certain culture characteristics. All urine samples were screened for the presence of ESBL genes ($bla_{CTX-M2}$, $bla_{CTX-M14}$, and $bla_{CTX-M15}$) with random-access quantitative PCR (qPCR) using the Cepheid GeneXpert ESBL-*ampC* prototype assay. The qPCR and microbiological culture results were compared. After the calculation of the sensitivity and specificity, discrepancies were investigated by whole-genome sequencing. In total, 276 urine samples were available for ESBL analysis (94 ESBL culture positive and 182 ESBL culture negative). The sensitivity and specificity for detection of ESBL genes were 90.4% and 98.4%, respectively. In nine samples (9.6%), no ESBL genes were detected with GeneXpert, while in the microbiological culture, an ESBL-positive organism was isolated. This was mainly explained by non-GeneXpert ESBL genes: $bla_{SHV}$-family ($n = 6$; 75.0%), $bla_{TEM}$-family ($n = 1$; 12.5%), and $bla_{SRT}$-family ($n = 1$; 12.5%). The positive and negative predictive values in a hypothetical clinical scenario with a 15% ESBL prevalence were 0.91 and 0.98, respectively. Regarding *ampC*, the specificity appears to be satisfactory, but the sensitivity is low. The Cepheid GeneXpert ESBL assay could be beneficial for the fast and accurate detection of ESBL genes in regions where the epidemiology of ESBL genes coincides with the targets in the panel.

**IMPORTANCE**   Early identification of complicated urinary tract infections caused by ESBL-producing Enterobacterales has the potential to limit the use of carbapenems to those patients without alternative antibiotic options and avoid the empirical use of carbapenems in patients without ESBL-producing bacteria. The purpose for such a test will differ by setting and ESBL prevalence rates. Countries with low ESBL rates and cephalosporins as empiric treatment (e.g., The Netherlands) will need a rule-in test to decide to use carbapenems, while countries with high ESBL rates and empiric carbapenem treatment will need a rule-out test for ESBLs to de-escalate therapy early. Anyway, such as a test would—at least theoretically—improve patient care and reduce selective pressure for the emergence of carbapenem resistance.

**KEYWORDS**   rapid diagnostics, urine samples, PCR, extended-spectrum beta-lactamase genes

Address correspondence to Sofie C. M. Tops, sofie.tops@radboudumc.nl.

This Radboudumc researcher-initiated study was supported in kind with free testing cartridges by Cepheid.

I.A.T. is an employee of Cepheid, and F.C.T. is a former Cepheid employee.

Antimicrobial resistance due to extended-spectrum beta-lactamase (ESBL) production by Enterobacterales is a serious health problem (1). The global intestinal ESBL carriage rate in healthcare settings increased from 7% in the period 2001–2005 to 25.7% in the period 2016–2020, whereas in the community, it increased 10-fold from 2.6% to 26.4%, respectively (2). Even in counties with historically low antimicrobial resistance rates like the Netherlands, the prevalence of ESBL-producing Enterobacterales has increased considerably from less than 1% in 1997 to 5% for *Escherichia coli* and to 8% for *Klebsiella pneumoniae* for inpatient departments excluding Intensive Care Units in 2021 (3, 4).

The increased prevalence of ESBL-producing Enterobacterales is problematic since it will likely lead to an increased number of patients with infections in whom empirical treatment with routine beta-lactam agents will be insufficient. In healthcare settings, this can result in higher mortality rates for multidrug-resistant infections compared to infections caused by non-ESBL infections (5), but it can also result in prolonged disease and/or disease progression for common infections such as complicated urinary tract infections (cUTIs). Currently, in the Netherlands, mainly third-generation cephalosporins are used for the empirical treatment of cUTIs. However, the increasing prevalence of ESBL-producing Enterobacterales in the Netherlands, as in many other countries, leads to increased use of carbapenems as empirical treatment for these infections (6). This trend may further drive antimicrobial resistance, particularly against last-resort antibiotics.

Early identification of cUTIs caused by ESBL-producing Enterobacterales has the potential to limit the use of carbapenems to those patients without alternative antibiotic options and avoid the empirical use of carbapenems in patients without ESBL-producing bacteria. The purpose for such a test will differ by setting and ESBL prevalence rates. Countries with low ESBL rates and cephalosporins as empiric treatment (e.g., The Netherlands) will need a rule-in test to decide to use carbapenems, while countries with high ESBL rates and empiric carbapenem treatment will need a rule-out test for ESBLs to de-escalate therapy early. Anyway, such as a test would—at least theoretically—improve patient care and reduce selective pressure for the emergence of carbapenem resistance. Since the standard methods of microbiological culture and antimicrobial susceptibility testing involve several overnight incubations, other more rapid techniques are required to guide empirical therapy at the time the patient is first evaluated. We investigated the diagnostic accuracy of the GeneXpert ESBL-*ampC* prototype assay cartridge for rapid detection of ESBL resistance genes directly in urine samples and assessed its potential impact of therapeutic decisions.

## MATERIALS AND METHODS

The study was performed between February and July 2021. All incoming urine samples from patients older than 18 years of age at two Dutch clinical medical microbiology laboratories (H1: an academic hospital and H2: a non-academic teaching hospital) were available for selection. In our study, we only used residual material from urine samples provided to our laboratory for routine diagnostics unless a patient explicitly objected against the use of residual material for research purposes using an opt-out method of consent. Therefore, based on national regulations, it was not necessary to obtain approval for our study from the Institutional Review Board.

Urine samples were selected according to a predefined list containing certain culture characteristics, i.e., microbial species, antimicrobial resistance profile, and urine clarity (Supplementary file I1). To optimally test the diagnostic accuracy of the GeneXpert ESBL-*ampC* prototype assay cartridge, we aimed for 100 ESBL culture-positive urine samples and heterogeneous selection of 200 ESBL culture-negative urine samples. In H2, only ESBL culture-positive urine samples were selected for study.

## Routine microbiological urine culture

10 µL of urine were inoculated on a Columbia III blood agar plate (H1: BD Diagnostic Systems, Sparks, MD, USA; H2: Thermo Scientific, Vienna, Austria) and a MacConkey agar plate without salt (H1: BD Diagnostic Systems, Sparks, MD, USA; H2: Thermo Scientific, Vienna, Austria). Plates were incubated for 18 up to 24 h at 36°C under $CO_2$ and $O_2$ conditions, respectively. Thereafter, agar plates were evaluated for the presence of UTI pathogens. Only clinically relevant bacteria that were present in sufficient quantities (at least $10^3$–$10^4$ micro-organisms per mL) as defined by the laboratory technician and/or clinical microbiologist were analyzed. The number of isolated bacterial species and the Gram stain also played a role in the assessment of whether or not a bacteria was considered clinically relevant. Bacteria were identified with matrix-assisted laser desorption ionization-time of flight mass spectrometry (MALDI-TOF MS, BioTyper library version 3, Bruker, Bremen, Germany). Antimicrobial susceptibility testing for each bacterial strain was performed using a broth-based microdilution test according to the manufacturer's instruction (H1: Phoenix system, M50, BD Diagnostic Systems, Sparks, MD, USA; H2: VITEK2, bioMérieux SA, Marcy l'Etoile, France).

For phenotypic ESBL confirmation, the EUCAST guidelines for the detection of resistance mechanisms, version 2.0 (published 11 July 2017; https://www.eucast.org), were used. Phenotypic ESBL confirmation was performed if the Minimal Inhibitory Concentration of at least one of the following antibiotics was >1 mg/L: cefotaxime, ceftriaxone, ceftazidime, or cefepime. The protocols for the detection of ESBL (and/or *ampC*) producing Enterobacterales are depicted in Supplementary file II.

## Multiplex qPCR testing using the GeneXpert

Testing was performed with random access multiplex qPCR using the GeneXpert ESBL-*ampC* prototype assay cartridge (Cepheid, Sunnyvale, USA) run on the GeneXpert GX-XVI instrument (Cepheid, Sunnyvale, USA). Selected urine samples were stored in the refrigerator and tested within 1 week after sample collection.

Urine samples were vortexed until homogeneity. 100 µL of urine was transferred into a 5 mL sample reagent bottle (Cepheid). Subsequently, the sample was vortexed for 10 seconds after which 1.7 mL was transferred into the GeneXpert ESBL-*ampC* prototype assay cartridge (Cepheid). The cartridge was placed into the GeneXpert instrument (GX-XVI, Cepheid) within 30 minutes, whereafter the assay was started according to the manufacturer's instructions. Urine samples were tested for the presence of the following ESBL resistance genes: $bla_{CTX-M2}$, $bla_{CTX-M14}$, and $bla_{CTX-M15}$. Besides the above-mentioned ESBL resistance genes, the ESBL-*ampC* prototype assay cartridge also contains primers to detect the *ampC* resistance genes $bla_{CMY-2}$, $bla_{DHA-1}$, and $bla_{FOX-5}$. Urine samples were reported as positive when a Ct value ≤35 was found for at least one of the resistance genes. The qPCR and routine microbiological culture results (considered as the reference method) were compared.

## Discrepancy analysis

In cases where the multiplex qPCR results differed from the results of the routine microbiological culture, additional analysis was performed to explain the discrepant results. In Supplementary file III, flowcharts on the processing of discrepant ESBL positive and negative urine samples can be found. For the ESBL culture-positive urine samples but with a GeneXpert negative result, first, all identified bacterial isolates with ESBL during routine microbiological culture were tested on GeneXpert. In case GeneXpert again reported a negative result, whole genome sequencing (WGS) was performed on the bacterial isolate to check for resistant genes using ResFinder 4.1 (https://cge.cbs.dtu.dk/services/ResFinder/). In the scenario that no (relevant) bacterial isolate(s) were retrieved after re-culturing, shotgun metagenomics were performed on the urine sample to check for microorganisms and ESBL resistance genes (Supplementary file IV). For ESBL culture-negative urine samples but with a GeneXpert positive result,

the urine sample was re-cultured to identify bacteria present in low (clinically non-relevant) quantities. In case additional bacterial isolates were detected after re-culturing, these were tested on GeneXpert. Moreover, the laboratory system was checked for ESBL culture-positive urine samples of the same patient within the previous month.

## Statistical analysis

Descriptive statistics were used as appropriate. To determine the accuracy of GeneXpert to detect the ESBL resistance genes present in the target panel, qPCR results were compared with the results of the routine microbiological culture (reference method). Sensitivity and specificity were calculated for the detection of ESBL. The discrepancy analysis for the detection of ESBL resistance genes was performed after the calculation of the sensitivity and specificity. For the statistical analysis, Excel was used (Microsoft Office Excel 2016 version 2002).

## RESULTS

### Diagnostic accuracy of the detection of ESBL resistance genes present in the panel

In total, 279 urine samples were selected for ESBL analysis. Three samples were left out of the analysis because of repeatedly invalid qPCR results. Therefore, 276 urine samples were included in the analysis, of which 94 urine samples were ESBL positive (34.1%) by routine microbiological culture. In 85 of these 94 ESBL culture, positive urine samples GeneXpert detected one or more ESBL resistance genes. The resistance genes detected were $bla_{CTX-M15}$ ($n = 54$; 63.5%) and $bla_{CTX-M14}$ ($n = 33$; 38.8%). In two urine samples, both $bla_{CTX-M14}$ and $bla_{CTX-M15}$ were detected. In addition, GeneXpert detected one or more ESBL resistance genes in 3 of the 182 ESBL culture-negative urine samples. The sensitivity and specificity for the detection of ESBL resistance genes are depicted in Table 1.

### Discrepancy analysis

#### False-negative qPCR results

In nine urine samples (9.6%), no ESBL resistance genes were detected with GeneXpert in contrast to the microbiological culture where an ESBL-positive organism was isolated. WGS showed that these discrepancies were explained by ESBL resistance genes that were not present in the target panel, i.e., $bla_{SHV}$-family ($n = 6$; 75.0%), $bla_{TEM}$-family ($n = 1$; 12.5%), and $bla_{SRT}$-family ($n = 1$; 12.5%). For one urine sample, no resistance gene were detected by WGS; therefore, for this sample, the discrepancy remained unresolved.

#### False-positive qPCR results

In three urine samples (1.6%), ESBL resistance genes were detected with GeneXpert in contrast to the microbiological culture, where no ESBL-positive organism was isolated.

**TABLE 1** Sensitivity and specificity of the GeneXpert ESBL-ampC prototype assay cartridge for detection of ESBL and *ampC* resistance genes directly from urine samples

| | GeneXpert positive | GeneXpert negative | Diagnostic accuracy |
|---|---|---|---|
| ESBL culture positive | 85 | 9 | Sensitivity: 85/94 = 90.4% 95% CI 82.6%–95.5% |
| ESBL culture negative | 3 | 179 | Specificity: 179/182 = 98.4% 95% CI 95.4%–99.7% |
| *ampC* culture positive[a] | 16 | 38 | Sensitivity: 16/54 = 29.6% 95% CI 18.0%–43.6% |
| *ampC* culture negative | 9 | 211 | Specificity: 9/220 = 95.9% 95% CI 92.4%–98.1% |

[a]Including the 45 urines containing group II Enterobacterales.

For two urines, the patient had an ESBL-positive urine culture within the previous month for which no antibiotic treatment was received. Therefore, we assumed that an ESBL-producing Enterobacterales was still present in (low quantities) the urine. In one urine, low quantities of ESBL-producing Enterobacterales were found after re-culturing of the urine sample.

## Diagnostic accuracy of the detection of *ampC* resistance genes present in the panel

In total, 274 urines were available for *ampC* analysis of which 45 urines contained group II Enterobacterales containing chromosomal *ampC* beta-lactamases. Phenotypic ampC confirmation was not performed for group II Enterobacterales. The GeneXpert was positive for 12 of these 45 urines (26.7%) with group II Enterobacterales (Table 2).

With regard to the remaining 229 urines, not containing Group II Enterobacterales, *ampC* was phenotypically detected in nine urine samples, which were confirmed by GeneXpert in four urine samples (44.4%). In addition, in 9 of the 229 urine samples not containing group II Enterobacterales, the GeneXpert was positive, but the urine sample was *ampC* culture negative. The *ampC* resistance genes detected were: $bla_{CMY-2}$ ($n = 6$; 24.0%), $bla_{DHA-1}$ ($n = 20$; 80%), and $bla_{FOX-5}$ ($n = 20$; 80%). The sensitivity and specificity can be found in Table 1. No discrepancy analysis was performed.

## Impact on therapeutic decisions

To investigate the diagnostic accuracy of the GeneXpert ESBL-*ampC* prototype assay cartridge for rapid detection of ESBL resistance genes, a specific subset of urines was selected. This subset doesn't accurately represent the general patient population. Therefore, we could only explore the hypothetical impact of using GeneXpert for the detection of ESBL resistance genes on therapeutic decision-making in Dutch daily clinical practice.

Positive predictive value (PPV) and negative predictive value (NPV) and the number of urine samples to screen with GeneXpert to detect one ESBL positive urine sample which would lead to a switch to (empiric) carbapenem antibiotics were calculated for different ESBL prevalence rates and the diagnostic accuracy of the assay in our study (Table 3). Our calculations (Table 3) support the use of the GeneXpert ESBL-*ampC* prototype assay and indicate that the assay is a promising tool in preventing the overuse of empirical last resort antimicrobial agents in a population with a high ESBL prevalence.

**TABLE 2** Detailed information about the discrepant urine samples

| Discrepancy | Micro-organisms isolated during routine microbiological culture | PCR result | Explanation of the discrepancy |
|---|---|---|---|
| ESBL culture positive and GeneXpert ESBL negative urine samples *n*= 9 | *Serratia marcescens* *E. coli* | Negative | WGS revealed *E. coli* with TEM-1 and TEM-52 resistance genes |
| | *K. pneumoniae* | Negative | WGS revealed *K. pneumoniae* with SHV-11 resistance gene |
| | *K. pneumoniae* | Negative | WGS revealed *K. pneumoniae* with SHV-11 and SHV-12 resistance genes |
| | *K. pneumoniae* | Negative | WGS revealed *K. pneumoniae* with SHV-11 and SHV-12 resistance genes |
| | *Serratia marcescens* | Negative | WGS revealed *Serratia marcescens* with SRT-1 resistance gene |
| | *K. pneumoniae* | Negative | WGS revealed *K. pneumoniae* with SHV-12, SHV-28, SHV-106 and SHV-187 resistance genes |
| | *K. pneumoniae* *Aerococcus urinae* | Negative | WGS revealed *K. pneumoniae* with SHV-187 resistance gene |
| | *E. coli* *K. pneumoniae* | Negative | Discrepancy not explained |
| | *K. pneumoniae* | Negative | WGS revealed *K. pneumoniae* with SHV-187 resistance gene |
| ESBL culture negative and GeneXpert positive urine samples *n* = 3 | Mixed bacterial flora | $bla_{CTX-M15}$ | Re-culturing detected low quantities of ESBL producing *E. coli* |
| | Mixed bacterial flora | $bla_{CTX-M15}$ | Patient had an ESBL positive urine culture within the previous month |
| | Mixed bacterial flora | $bla_{CTX-M15}$ | Patient had an ESBL positive urine culture within the previous month |

**TABLE 3** Positive predictive value, negative predictive value, and number needed to screen for different ESBL prevalence rates

| ESBL prevalence (%) | Positive predictive value % (95% CI) | Negative predictive value % (95% CI) | Number needed to screen |
|---|---|---|---|
| 5 | 74.3 (48.4–89.9) | 99.5 (99.1–99.7) | 22 |
| 10 | 85.9 (66.5–94.9) | 98.9 (98.0–99.4) | 11 |
| 15 | 90.6 (75.9–96.8) | 98.3 (96.9%–99.1%) | 7 |
| 20 | 93.2 (81.7–97.7) | 97.6 (95.7–98.7) | 6 |
| 25 | 94.8 (85.6–98.3) | 96.9 (94.3–98.3) | 4 |
| 30 | 95.9 (88.4–98.6) | 96.0 (92.8–97.8) | 4 |
| 35 | 96.7 (90.6–98.9) | 95.0 (91.1–97.3) | 3 |
| 40 | 97.3 (92.2–99.1) | 93.9 (92.0–97.4) | 3 |

## DISCUSSION

The GeneXpert ESBL-*ampC* prototype assay is a rapid, random-access, cartridge-based assay with a turnaround time of 75 minutes. We showed that the Cepheid ESBL-*ampC* prototype assay is suitable for the detection of ESBL resistance genes directly in urine samples. In clinical practice, the assay could be used for fast and accurate detection of ESBL genes in patients suspected of cUTI to guide empirical antimicrobial treatment, while waiting for the results of microbiological culture. In this way, the use of carbapenems can be limited to those patients without alternative antimicrobial options, thus avoiding the use of carbapenems in patients without ESBL-producing bacteria. Using PCR to identify resistance genes to guide antimicrobial treatment is common for distinguishing methicillin-resistant *Staphylococcus aureus* from methicillin-susceptible *S. aureus* in blood cultures using the *mecA* gene (7), for detecting mutations associated with rifampin resistance in *Mycobacterium tuberculosis* (8), and as part of multiple syndromic panels (9). This approach can improve patient care and reduce the selective pressure that leads to further development of antimicrobial resistance.

There are several factors to consider for potential future use of this assay in clinical practice. First, the multiplex qPCR appears to be very sensitive and detected ESBL genes from Enterobacterales that were present in very low, potentially non-clinically relevant quantities ($n = 3$). This could lead to the unnecessary use of carbapenems in patients with low bacterial burdens in the urinary tract. However, in symptomatic patients, low bacterial counts could be due to prior antimicrobial treatment. Cycle threshold values to maximize the sensitivity and specificity of the assay have yet to be finalized. Second, multiplex qPCR detects the presence of an ESBL genotype but does not indicate that the gene is expressed resulting in an ESBL phenotype in the bacterial host species. Again, this could lead to the unnecessary use of empirical carbapenems. However, we observed a high concordance of genotype and phenotypes in the urine samples tested in this study. Last, it should be noted that the current assay is limited by the number of targeted ESBL resistance genes ($bla_{CTX-M2}$, $bla_{CTX-M14}$, and $bla_{CTX-M15}$). In fact, the prevalence of ESBL-specific resistance genes may be dependent on the geographical regions, which may affect the sensitivity and clinical value of the test. Previous studies reported that the ESBL resistance genes included in the target panel are the most prevalent ones in the Netherlands (10–12). However, six specimens with organisms containing $bla_{SHV}$ ESBLs were detected by WGS, while one resistance gene ($bla_{CTX-M2}$) in the target panel was not present in the selected urine samples. For the above-mentioned reasons, the value of the GeneXpert ESBL/*ampC* prototype assay will depend on the prevalence of ESBL genes in the setting where the test is applied. qPCR should not replace routine microbiological cultures and should be considered as a supplementary test to guide the initial selection of antimicrobial therapy. Antimicrobial susceptibility testing is still needed ultimately to adjust treatment and ideally move to narrow-spectrum therapy.

To evaluate the sensitivity of the Cepheid ESBL-*ampC* prototype assay for the detection of *ampC* resistance genes, a larger set of AmpC positive samples will be necessary. Based on our results, the specificity appears to be satisfactory, but the sensitivity is low. In the Netherlands, only 2.4% of the patients were identified as rectal

carriers of *ampC* β-lactamase-producing *E. coli*, and in contrast to ESBL carriage which is increasing, the number of patients with *ampC* β-lactamase-producing *E. coli* is declining (13). The clinical relevance of a PCR test that detects plasmid-mediated *ampC* genes has yet to be determined. Because of the low clinical relevance, the discrepancy with regard to *ampC* was not further analyzed.

In conclusion, we demonstrated that the Cepheid ESBL-*ampC* prototype assay has acceptable diagnostic accuracy for the rapid detection of ESBL resistance genes directly in human urine samples. The assay is a promising tool in preventing the overuse of empirical last resort antimicrobial agents in a population with a high ESBL prevalence (15% or more). Before seeking regulatory registration for using the assay in clinical practice, technical validation of the assay is necessary, and a clinical study is needed to assess the impact of such a strategy. For *ampC* resistance genes, the clinical need for early detection of *ampC* resistance genes with qPCR is lower, and the significance of the qPCR result for clinical practice is more complex.

## AUTHOR AFFILIATIONS

[1]Department of Medical Microbiology and Radboudumc Center for Infectious Diseases, Radboud University Medical Center, Nijmegen, The Netherlands
[2]College of Arts and Sciences, University of Dayton, Dayton, Ohio, USA
[3]Cepheid, Sunnyvale, California, USA

## AUTHOR ORCIDs

Sofie C. M. Tops ⓘ http://orcid.org/0000-0003-2492-6943
Jordy P. M. Coolen ⓘ http://orcid.org/0000-0003-0443-6250
Willem J. G. Melchers ⓘ http://orcid.org/0000-0002-5446-2230

## AUTHOR CONTRIBUTIONS

Sofie C. M. Tops, Data curation, Formal analysis, Methodology, Supervision, Writing – original draft | Claire E. P. Schapendonk, Investigation, Methodology, Writing – original draft | Jordy P. M. Coolen, Formal analysis, Investigation, Writing – review and editing | Fred C. Tenover, Methodology, Resources, Writing – review and editing | Isabella A. Tickler, Methodology, Resources, Writing – review and editing | Willem J. G. Melchers, Methodology, Supervision, Writing – review and editing | Heiman F. L. Wertheim, Conceptualization, Funding acquisition, Methodology, Supervision, Writing – review and editing

## DATA AVAILABILITY

Data was made publicly available upon publication. DOI: 10.17026/dans-x2f-6fdq.

## ADDITIONAL FILES

The following material is available online.

### Supplemental Material

**Supplementary appendix (Spectrum03116-23-s0001.docx).** Supplementary file I: Selection criteria for urine samples (planned and actual numbers). Supplementary file II: Protocol of ESBL and AmpC detection in Enterobacterales. Supplementary file III: Flow charts for processing discrepant ESBL and/or AmpC positive urine specimen and ESBL and AmpC negative urine specimen. Supplementary file IV: Whole genome sequencing methodology.

## Open Peer Review

**PEER REVIEW HISTORY (review-history.pdf).** An accounting of the reviewer comments and feedback.

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
