## [Reviewer comments · Microbiology Spectrum]

Microbiology Spectrum

The diagnostic accuracy of the GeneXpert ESBL-ampC prototype assay for rapid PCR-based detection of Extended-Spectrum Beta-Lactamase genes directly from urine

Sofie C.M. Tops, Claire E.P. Schapendonk, Jordy Coolen, Fred C. Tenover, Isabella Tickler, Willem Melchers, and Heiman Wertheim

Corresponding Author(s): Sofie C.M. Tops, Radboudumc

Review Timeline:

Submission Date:

September 18, 2023

Accepted:

October 2, 2023

Editor: Cezar Khursigara

Reviewer(s): The reviewers have opted to remain anonymous.

Transaction Report:

DOI: <https://doi.org/10.1128/spectrum.03116-23>

October 2, 2023

Mrs. Sofie C.M. Tops
Radboudumc
Medical Microbiology
Geert Grooteplein Zuid 10
Nijmegen 6525 GA
Netherlands

Re: Spectrum03116-23 (The diagnostic accuracy of the GeneXpert ESBL-ampC prototype assay for rapid PCR-based detection of Extended-Spectrum Beta-Lactamase genes directly from urine)

Dear Mrs. Sofie C.M. Tops:

Your manuscript has been accepted, and I am forwarding it to the ASM Journals Department for publication. You will be notified when your proofs are ready to be viewed.

Sincerely,

Cezar Khursigara
Editor, Microbiology Spectrum
